# Development of Stable Liposomal Drug Delivery System of Thymoquinone and Its In Vitro Anticancer Studies Using Breast Cancer and Cervical Cancer Cell Lines

**DOI:** 10.3390/molecules27196744

**Published:** 2022-10-10

**Authors:** Mohammad Hossain Shariare, Md Asaduzzaman Khan, Abdullah Al-Masum, Junayet Hossain Khan, Jamal Uddin, Mohsin Kazi

**Affiliations:** 1Department of Pharmaceutical Sciences, North South University, Dhaka 1229, Bangladesh; 2Research Center for Preclinical Medicine, Southwest Medical University, Luzhou 646000, China; 3Center for Nanotechnology, Department of Natural Sciences, Coppin State University, Baltimore, MD 21216, USA; 4Department of Pharmaceutics, College of Pharmacy, King Saud University, P.O. Box 2457, Riyadh 11451, Saudi Arabia

**Keywords:** liposome, thymoquinone, cell cytotoxicity, particle size, anticancer

## Abstract

Thymoquinone, a well-known phytoconstituent derived from the seeds of *Nigella sativa*, exhibits unique pharmacological activities However, despite the various medicinal properties of thymoquinone, its administration in vivo remains challenging due to poor aqueous solubility, bioavailability, and stability. Therefore, an advanced drugdelivery system is required to improve the therapeutic outcome of thymoquinone by enhancing its solubility and stability in biological systems. Therefore, this study is mainly focused on preparing thymoquinone-loaded liposomes to improve its physicochemical stability in gastric media and its performance in different cancer cell line studies. Liposomes were prepared using phospholipid extracted from egg yolk. The liposomal nano preparations were evaluated in terms of hydrodynamic diameter, zeta potential, microscopic analysis, and entrapment efficiency. Cell-viability measurements were conducted using breast and cervical cancer cell lines. Optimized liposomal preparation exhibited polygonal, globule-like shape with a hydrodynamic diameter of less than 260 nm, PDI of 0.6, and zeta potential values of −23.0 mV. Solid-state characterizations performed using DSC and XRPD showed that the freeze-dried liposomal preparations were amorphous in nature. Gastric pH stability data showed no physical changes (precipitation, degradation) or significant growth in the average size of blank and thymoquinone-loaded liposomes after 24 h. Cell line studies exhibited better performance for thymoquinone-loaded liposomal drug delivery system compared with the thymoquinone-only solution; this finding can play a critical role in improving breast and cervical cancer treatment management.

## 1. Introduction

In many cases, solubility or other bio-incompatibility issues between drug molecules and biological systems lead to failure in delivering the drug molecules to the target site and, thus, failed to improve the therapeutic outcome. Therefore, loading of drugs to a suitable carrier may lead to their target-specific delivery to tissue or cells, which can be an effective approach to overcome these hurdles [1,2]. The encapsulation of phytochemicals in nanocarriers safeguards them by avoiding their exposure to the digestive tract environment as well as helping in targeting the desired site of action using suitable ligands. Several technologies, such as nanotechnology, have been utilized for improving these incompatibilities. Nanotechnology in drug delivery can improve the encapsulation of drugs which may improve their solubility, permeability, and stability. Nano-drug-delivery systems can deliver drug to the specific site of action and provide better efficacy with fewer side effects [3]. Nanotechnology allows atomic- and molecular-level fabrication which helps in designing, optimizing, producing, and characterizing nanocarriers [4,5]. The small sizes of nanocarriers make them unique as potential agents for biomedical applications [1,3,4,6]. Different types of nanocarriers, such as liposomes [7], silica nanoparticles [8], and metal-based and carbon-based nanomaterials [9,10], have been tested in biomedical fields for drug delivery, including for the delivery of oridonin and methotrexate [11], of paclitaxel [12], and of small or short interfering RNA (SiRNA) [13].

Phytochemicals have been used in medical practice from ancient times to the present for their promising therapeutic outcomes in treating different diseases [14]. Thymoquinone is a natural phytoconstituent of *Nigella sativa* seed oil [15]. In 1963, El-Dakhakhny first isolated thymoquinone (TQ) from *Nigella sativa* seeds using a chromatographic system [16]. There is an enormous body of research on the beneficial biological effects of TQ, including antioxidant [17], anti-inflammatory [18], antidiabetic [19], antinociceptive [20], nephroprotective [21,22], neuroprotective [22,23], chemoprotective, and chemo-curative characteristics [24]. Many types of carcinogenic cell proliferation were reported to be suppressed by TQ [25,26,27,28,29,30], with very little toxicity for normal cells [31]. Although TQ possesses many beneficial biological effects, its physicochemical properties (poor solubility, bioavailability, and stability) are obstacles to the absorption and distribution of TQ in biological systems [32,33]. Oral administration can cause chemical and enzymatic degradation of TQ in the gastrointestinal tract [4]; TQ also exhibits sensitivity under different pH, light, and temperature conditions [34].

Previous research in this area has showed that polymeric, SLN, and lipid-based nano preparation of TQ can improve its solubility and bioavailability for the treatment of breast cancer cell lines. Thymoquinone and its nano preparation also showed marked efficacy against liver cirrhosis, diabetes, inflammation, CNS diseases, and hepatotoxicity due to its strong antioxidant properties [18,35,36,37]. In these studies, nano preparations of TQ were characterized in terms of particle size distribution, zeta potential, entrapment efficiency, and accelerated stability study. However, to the best of our knowledge, no studies have been reported on the gastric pH stability study of liposomal preparation of TQ. Previous cell line studies of TQ and its nano preparation have mostly been performed on breast cancer and colon cancer cell lines [38,39]; no reports have been published so far on the effect of TQ nano preparation on cervical cancer cell lines. Therefore, the primary focus of this study is to develop a liposome-based nano drug delivery system for delivering TQ to improve its stability in gastric media with low average particle size and high entrapment efficiency. The liposomal nano drug delivery system of TQ was characterized using TEM, Malvern Zetasizer, XRD, and DSC. In vitro cancer cell line (breast cancer and cervical cancer) studies were performed for assessing the liposomal nano drug delivery system of thymoquinone (TQ).

## 2. Results

### 2.1. TEM Analysis

Transmission electron microscopic (TEM) images (Figure 1) were used to examine the surface morphology and globule size; results showed that, liposomal preparation of thymoquinone were in thesize range of <250 nm and polygonal-like shapes (Figure 1B). 

Particle size distributions of <250 nm were also observed when characterized using the DLS method (Table 1). Polygonal-shape-like morphology for liposomes were also observed by Johnsson and Edwards (2003) [40] using cryo TEM and Rushmi et al. (2017) [41]. Post-dilution of the liposomal formulation with deionized water (1 in 100 dilution), showed that the mixture was well-dispersed without any particle aggregation.

### 2.2. DLS Analysis

Particle size distribution and zeta potential value of liposomal batches of TQ were determined to evaluate average particle size and stability of liposomal preparations by the DLS (Malvern Zetasizer, Malvern Instruments, Malvern, UK) method. The average sizes of liposomes prepared in this study were in the range of 170–250 nm (Table 1). Poor aqueous solubility led to low oral bioavailability, which is frequently observed with BCS class II drugs, including thymoquinone. Therefore, a better drug-delivery system is required if we are to enhance the solubility and bioavailability of thymoquinone for effective therapeutic outcomes. Liposomal drug-delivery systems can improve the dissolution, solubility, and stability of the drug, protect drugs from degradation, and improve their bioavailability, leading to better therapeutic outcomes.

Liposomal formulation 1 (F1) of thymoquinone, prepared without cholesterol, exhibited low average particle size and PDI compared with liposomal formulation 2 (F2) (Table 1). High entrapment efficiency and zeta potential value was observed in liposomal formulation 2 (F2) of thymoquinone. The high entrapment efficiency, zeta potential, and average size of liposomal formulation F2 were possibly related to the use of cholesterol for this batch, which was also observed in a previous research study [41]. Therefore, liposomal formulation F2 was used in gastric pH stability study and cancer cell line studies.

### 2.3. DSC Analysis

DSC thermograms of raw thymoquinone and its liposomal preparations are shown in Figure 2. A sharp endothermic peak of pure thymoquinone was observed at around 47 °C due to its melting temperature. However, no sharp peak was observed for the liposomal preparation of thymoquinone. The absence of an endothermic peak for liposomal preparation of thymoquinone indicates that the conversion from a crystalline state to an amorphous state occurred during the liposomal preparation (Figure 2) or the freeze-drying process in the liposomal preparation of TQ. This phenomenon might also be related to the drug being molecularly dispersed in the liposomal system which was used during the DSC study.

### 2.4. XRD Analysis

XRD scan (Rigaku Corporation, Tokyo, Japan) was performed to investigate the solid-state form of pure thymoquinone and its liposomal freeze-dried preparation. Figure 3 shows the XRD spectra of pure thymoquinone and its liposomal preparation. Pure thymoquinone yielded a sharp peak in the spectra (Figure 3A), while no sharp peaks were observed for the liposomal preparation of thymoquinone (Figure 3B). The disappearance of a sharp peak in the thymoquinone-loaded liposomal preparation was related to the conversion of the drug into its amorphous form, the liposomal preparation of TQ; this probably occurred during the preparation or freeze-drying process used to analyze the liposomal preparation using XRD.

### 2.5. In Vitro Cell Line Study of Thymoquinone and Its Liposomal Preparation

The MTT assay results for four different cell lines treated with pure thymoquinone and the thymoquinone-loaded liposomal preparation (F2) are presented in Figure 4. The cell cytotoxicity analysis showed that TQ reduced the percentage of living cells in a dose-dependent manner, i.e., induced cytotoxicity. Similarly, liposomal preparation of TQ induced cytotoxicity in a dose-dependent manner in all four cell lines (two breast cancer and two cervical cancer). Interestingly, liposomal preparation of TQ was found to be more efficient than pure TQ and showed increased cytotoxicity (*n* = 3) (Figure 4). This can be furthered confirmed by the IC50 values of pure TQ compared with liposomal batch F2 (Table 2).

This phenomenon is probably related to the increased solubility, permeability, bioavailability, and stability and reduced metabolic cytotoxicity of thymoquinone when delivered through a liposomal nano-delivery system; this was observed by previous researchers using different anticancer drugs [38].

### 2.6. Stability of Liposomal Preparation of Thymoquinone in Gastric pH

The precipitation was confirmed by the appearance and droplet size analysis of the batch at various intervals (0, 2, 4, and 24 h; see Table 3).

Thymoquinone-free liposomes (blank liposomes) and thymoquinone-encapsulated liposomes were diluted with gastric media (pH 1.2) to investigate the stability of liposomes in gastric conditions. The physical appearances of the liposomal dispersions (Figure 5) were then evaluated, and no precipitation was observed in the thymoquinone-loaded representative batch. The DLS particle size analysis data showed no significant difference for the batch from 254.31 nm (diluted with water) to 255.45 nm (diluted with gastric media) at 0 h. The overall data from the stability study in gastric pH suggest that no significant physical changes, such as the precipitation or degradation of the TQ-loaded liposomes, were recorded even after 24 h (Table 3).

There was no significant change observed within 24 h of the dispersion (particle size increased slightly up to 270.33 nm) in gastric media (Figure 5). The dispersion image at 24 h showed no significant changes in physical appearance for liposomal formulation (Figure 5). However, the appearance was slightly clearer than the initial appearance. The objective of this test was to observe whether any precipitation or phase separation occurred over time. These results suggest that the thymoquinone-loaded liposomal formulation produced nanoemulsion in gastric pH and stabilized sufficiently for absorption to occur. The gastric pH stability study result showed that no physical changes (precipitation or degradation) were observed for the thymoquinone-loaded batch or the blank liposome batch even after 24 h.

With reference to the lipophilic phytoconstituents, their indigents’ water solubility restricts their assimilation by the digestive tract, which results in poor bioavailability. Similarly hydrophilic molecules have to endure various conditions (enzyme, ionic strength, pH, etc.) in the digestive tract, which results their decomposition [42].

## 3. Discussion

Liposomes are mainly composed of phospholipids of either natural or synthetic origin. Liposomes offer the advantages of encapsulating different phytoconstituents, targeting, masking taste, and stimuli-sensitive delivery. The encapsulation of TQ in liposomes using cholesterol (batch F2) can offer several advantages associated with the liposomal carrier system, such as low toxicity, biocompatibility, small size, high stability, ease of delivery, and biodegradability. A study [41] conducted by Rushmi et al. showed that the use of liposomes as drug carriers could increase the drug uptake, while degradation rates could be decreased. Liposomes have been used to protect drug molecules (both hydrophilic and hydrophobic) from the drastic conditions of low pH when administered orally. In addition, it has been found to increase drug transport into the intestinal lymph which thus led to increased systemic bioavailability [43].

In addition, previous stability studies documented stable liposomal preparations in a temperature range of 2–8 °C. However, some of these liposomal formulations were stable at 25 °C [41]. This is probably due to the differences in the phospholipid/cholesterol composition, content or concentration in formulation. This phenomenon is further proved in the current studies as TQ-loaded formulation was stable under stomach pH conditions and temperatures for several hours.

The results of the microscopic and DLS methods suggest that the liposomal preparations of thymoquinone were nanosized. TEM data showed possible polygonal-like morphology for the thymoquinone-loaded nano preparation, which was also observed by Johnsson and Edwards (2003), who mentioned a liposome-like structure [40]. This study also showed high average particle size, PDI, zeta potential, and entrapment efficiency for liposomal batch F2, which was prepared using cholesterol. Previous research suggests that cholesterol can possibly minimize the pores of liposome bi-layer vesicles and give a more confined structure; such changes enhance the entrapment efficiencies. The increased zeta potential value of batch F2, made using cholesterol, was probably related to the increase in electric potential of the electrical double layer of the liposomal suspension of TQ (F2). The nano liposomal batch prepared using cholesterol (F2) also exhibited large average size, which was also observed in previous research studies [41]. Solid-state characterization data (DSC and XRD) suggest that a liposomal nano preparation of TQ may be converted into an amorphous form possibly during the freeze-drying process before characterization or during the preparation of nano liposomal TQ. Cancer cell line studies suggest that both TQ and its liposomal preparation induced cytotoxicity for breast cancer and cervical cancer cell line studies in a dose-dependent manner. Results also showed that the liposomal nano preparation of TQ was found to be more efficient than the TQ-only preparation and showed increased cytotoxicity. This may be due to the improved permeability, bioavailability, and stability of TQ in the liposomal nanopreparation. Gastric pH stability data showed no significant physical changes for blank or TQ-loaded liposomal nanopreparation after 24 h, which is probably due to the formation of nanoemulsion for these preparations in gastric pH. Nanoemulsion formation has improved the stability of TQ in acidic environments and gave sufficient time for absorption; this led to a better cytotoxic effect against different cancer cell lines compared with TQ alone.

## 4. Materials and Methods

### 4.1. Materials

Thymoquinone (TQ: purity 98.6%) and cholesterol were obtained from Sigma-Aldrich (Frankfurt, Germany). All the solvents used in this study were HPLC-grade and purchased from Sigma-Aldrich (Nagpur, India). Egg phospholipid was prepared in-house [32] (Pharmaceutical Science Lab, North South University, Dhaka, Bangladesh).

### 4.2. Preparation of Liposomes

Liposomal batches of thymoquinone were prepared using the ethanol-injection method. Two different preparations of liposomes were developed using thymoquinone: F1 and F2 (Table 4. At first, the thymoquinone, phospholipid, and cholesterol were dissolved in 10 mL ethanol at 500 rpm for 10 min. The phospholipid solution containing thymoquinone (10 mL) was then added into deionized water (10 mL) at 1 mL/s and stirred at 1000 rpm for 15 min. Liposomal preparation of thymoquinone then heated at 30 °C until the volume came down to 10 mL.

### 4.3. Transmission Electron Microscopy (TEM) Analysis

Microscopic characterization of the thymoquinone preparation was performed using a JEOL JEM1010 transmission electron microscope ((JEOL Ltd., Akishima, Tokyo, Japan). The sample was freshly prepared, placed on a carbon-coated copper grid, and the lipid components were stained with osmium. The dried sample was loaded into the TEM and viewed at different magnification levels.

### 4.4. Particle Size and Zeta Potential Determination

A Malvern Zetasizer Nano ZS90, Malvern Instruments, Malvern, UK, was used to determine the average particle size, polydispersity, and zeta potential values for the liposomal batches of thymoquinone at 25 °C. The analysis was performed in triplicate, and the average value was used.

### 4.5. Entrapment Efficiency Determination of Thymoquinone

The supernatant, containing free drug, was separated from the liposomal suspension using centrifugation at around 9700 rcf for 1 h at 4 °C. Then, the entrapment efficiency of thymoquinone for the liposome batches was determined after dissolving liposomes with ethanol, followed by sonication for 10 min [41]. The solution was diluted with deionized water and the absorbance of the encapsulated thymoquinone in the diluted solution was determined at 255 nm using a UV-Vis spectrophotometer. The concentration of the encapsulated and free drug in the supernatant was determined using a calibration curve in triplicate. The encapsulation efficiency was calculated using the following formula:(1)$$\mathrm{Encapsulation}\;\mathrm{efficiency}(\%)=\frac{\mathrm{Entrapped}\;\mathrm{drug}}{\mathrm{Total}\;\mathrm{drug}}\times 100$$

### 4.6. X-ray Diffraction (XRD)

X-ray diffraction (XRD) was used to characterize the solid-state form of thymoquinone, and the liposomal preparation of thymoquinone was freeze-dried using a Rigaku multiflex diffractometer (Rigaku Corporation, Tokyo, Japan) at a wavelength of 1.5418 A. The analyses were scanned over a 2θ range of 3–60° at a rate of 0.50°/min and scan step time of 1.0 s.

### 4.7. Differential Scanning Calorimetry (DSC)

The thermal behavior of the thymoquinone and the freeze-dried liposomal thymoquinone nanopreparation samples were characterized by DSC (DSC-60, Shimadzu, Kyoto, Japan). Samples (approximately 5 mg) were taken into a sealed aluminum sample pan and scanned at a heating rate of 10 °C/min over a temperature range of 50–250 °C. The temperature scale was calibrated using a pure-indium standard (melting point of 156.6 °C).

### 4.8. Cancer Cell Line Study

Two breast cancer cells lines (BT-549 and MDA-MD-231) and two cervical cancer cell lines (SiHa and HeLa) were cultured in DMEM culture media (Gibco, Thermo Fisher ScientificTM, Beijing, China), supplemented with 10% fetal bovine serum (FBS) (Gibco, Life Technologies, NSW, Australia) in cell culture dishes or flasks at 37 °C with 5%CO^2^. All cell lines were obtained from American Type Culture Collection (ATCC, Manassas, VA, USA): BT549 (ATCC# HTB-122™), MDA-MB-231 (ATCC#HTB-26™), SiHa (ATCC#HTB-35™), and HeLa (ATCC#CCL-2™). The stock concentrations of standard drug thymoquinone (TQ) were prepared in dimethyl sulfoxide (DMSO).

Cell viability or cytotoxicity was assayed using the Cell Counting Kit-8 (CCK-8) (Beyotime, Nanjing, China). Cells (5 × 10^4^ cells/mL) were cultured in a 96-well cell-culture plate, and they were treated with 1–50 µM of TQ and nanocoated TQ (TQ-nano) for 48 h. DMSO was used as control where 0 µM concentration is used as the blank liposome/no-drug treatment. CCK-8 reagent (10 µL) was added into each well and kept at room temperature for 1 h at the end of the incubation periods. The absorbance was determined at 450 nm in a microplate spectrophotometer (Multiskan™ GO, Thermo Scientific, Vantaa, Finland). The colour intensity represented the percentage of living cells.

### 4.9. Stability of Liposomal Formulation in Gastric pH

The stability study was performed using the F2 liposomal batch of thymoquinone under a stomach environment at room temperature. After dilution (1 in 1000 dilution, 1 mg/mL) of TQ-loaded representative liposomes with gastric media (pH 1.2), the appearance of the dispersion was visually assessed to investigate the precipitation of TQ in the gut.

## 5. Conclusions

DLS and microscopic data suggest that the liposomal preparations of thymoquinone were <250 nm in size. Liposomal formulations prepared using cholesterol exhibit high entrapment efficiency; solid-state characterization data demonstrate that the liposomal nano preparation was amorphous in nature. The in vitro cell line data showed that the liposomal preparation of thymoquinone was more efficient compared with thymoquinone alone. Stability study data demonstrated that liposomal preparation of thymoquinone was physico-chemically stable in both water and gastric pH (1.2). Therefore, this study indicates that liposomal preparation of thymoquinone can be developed to improve the in vivo performance and physico-chemical stability of hydrophobic, poorly soluble, and unstable ingredients like thymoquinone and those similar to it.

## Figures and Tables

**Figure 1 molecules-27-06744-f001:**
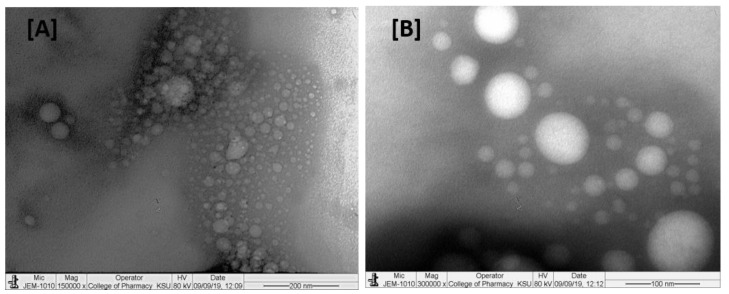
TEM images of the thymoquinone-loaded representative liposomal formulation. Images (**A**,**B**) represent the droplet sizes with 200 nm and 100 nm scales, respectively.

**Figure 2 molecules-27-06744-f002:**
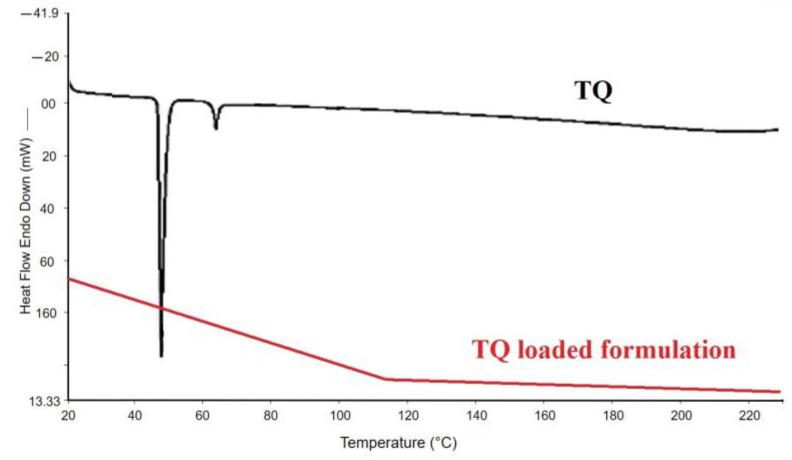
DSC curves of pure thymoquinone (TQ) and freeze-dried TQ-loaded liposomal formulation (batch 2).

**Figure 3 molecules-27-06744-f003:**
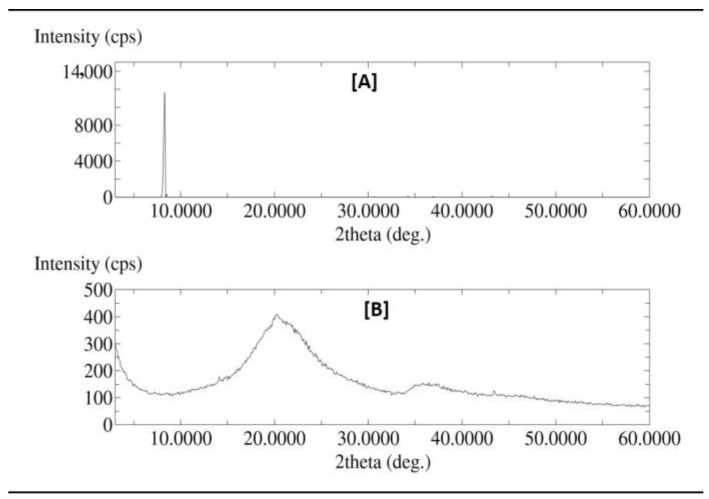
XRD of (**A**) pure thymoquinone and (**B**) thymoquinone-loaded liposomal formulation (F2).

**Figure 4 molecules-27-06744-f004:**
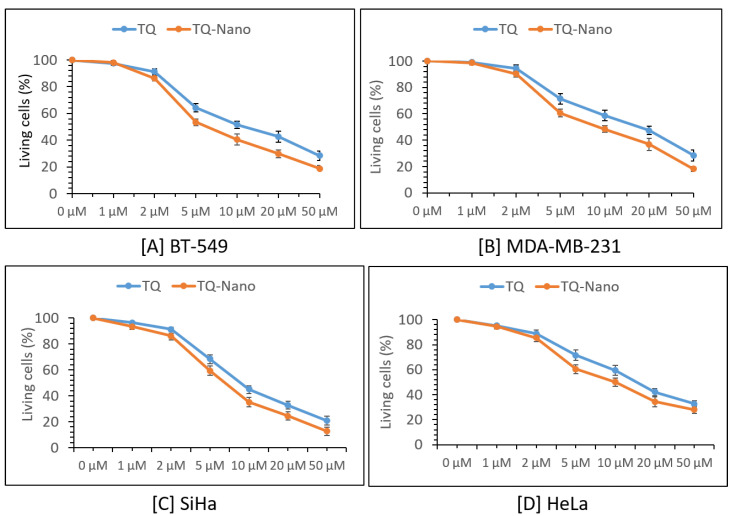
Effect of liposomal preparation of thymoquinone’s cytotoxic effect on cancer cell lines. Both TQ and liposomal preparation of TQ (TQ-nano) showed dose-dependent activity against BT-549 and MDA-MB-231 breast cancer cells, and SiHa and HeLa cervical cancer cells; however, in every case, the liposomal preparation of TQ increased cytotoxicity. Results are shown as mean ± SD (*n* = 3).

**Figure 5 molecules-27-06744-f005:**
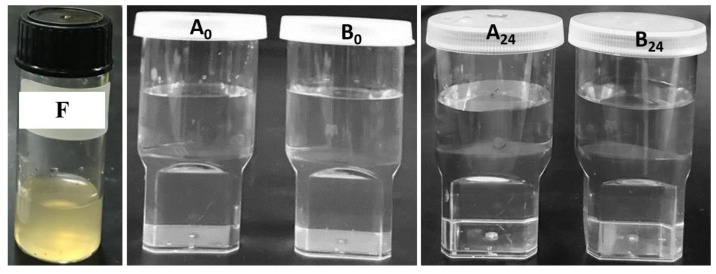
Liposomal formulation (F2) before dilution [F] and after dilution (1 in 1000 dilution, 1 mg/mL): A_0_ and A_24_ with water at 0 h and 24 h, and B_0_ and B_24_ with gastric pH (1.2 pH) at 0 h and 24 h, respectively.

**Table 1 molecules-27-06744-t001:** Average particle size, polydispersity index, zeta potential, and entrapment efficiency of thymoquinone liposomal preparations (data are presented as mean ± standard error of mean, where *n* = 5).

No	Average Size (nm)	PDI	Zeta Potential (mV)	Entrapment Efficiency %
F1	174.72 ± 11.40	0.312 ± 0.056	−13.3 ± 3.21	73.4% ± 1.6
F2	254.31 ± 8.32	0.604 ± 0.023	−23.0 ± 4.24	87.1% ± 2.3

**Table 2 molecules-27-06744-t002:** IC 50 value of the representative liposomal formulation on different cancer cell lines.

Cell Line	Name of Formulation	IC_50_
BT549	TQ	23.5970
	Nano-TQ	15.7481
MDA-MB-231	TQ	26.3413
	Nano-TQ	19.1378
SiHa	TQ	19.6475
	Nano-TQ	14.8046
HeLa	TQ	26.9797
	Nano-TQ	21.1733

**Table 3 molecules-27-06744-t003:** Particle sizes of the representative F2 batch after dilution in acidic/gastric solution (pH-1.2) (1 in 1000 dilution) (data are presented as mean ± standard error of mean, where *n* = 5).

Batch- (F2)	Particle Size	PDI
0 h	255.45 ± 8.76	0.313 ± 0.03
2 h	253.60 ± 12.26	0.238 ± 0.01
4 h	262.23 ± 4.65	0.644 ± 0.04
24 h	270.33 ± 22.42	0.546 ± 0.03

**Table 4 molecules-27-06744-t004:** Formulation attributes used for the preparation of liposomal drug-delivery system of thymoquinone.

Batch	Egg Phospholipid (mg)	Cholesterol (mg)	Thymoquinone (mg)	Stirring Rate (rpm) during Mixing of Ethanol and Water
F1	20	-	10	1000
F2	20	4	10	1000

## Data Availability

Not applicable.

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
