# Peer review of "Development of Stable Liposomal Drug Delivery System of Thymoquinone and Its In Vitro Anticancer Studies Using Breast Cancer and Cervical Cancer Cell Lines"

_molecules, 2022, doi:10.3390/molecules27196744_

Round 1

Reviewer 1 Report (Previous Reviewer 1)

The authors have made adequte revision and the manuscript can be accepted at its present form.

Reviewer 2 Report (Previous Reviewer 2)

Accepted 

Reviewer 3 Report (Previous Reviewer 3)

The manuscript looks much better.

All neccesary changes have been made. 

So I recommended this manuscript to publication in Molecules. 

This manuscript is a resubmission of an earlier submission. The following is a list of the peer review reports and author responses from that submission.

Round 1

Reviewer 1 Report

This manuscript report a thymoquinone loaded nanoparticle for in vitro cancer therapy. The results showed good anticancer efficacy of the nanoparticles. However, some issues should be addressed to improve the quality of this manuscript. I recommend major revision.

1. Have the authors tried other kinds of materials to prepare thymoquinone nanoparticles? The nanoparticles reported in the manuscript have large size (nearly 200 nm), as small nanoparticles may have a better stability and anticancer efficacy.

2. According to the TEM images, the size of nanoparticles were not uniform. The size of large ones was several fold higher than that of the small ones. Can the authors improve the preparation process to give nanoparticles with more uniform sizes?

3. In caption of Figure 5, the statement "Liposomal formulation (F2) in anhydrous form [F]" was not precise. The formulation [F] shown in the figure is solution, not anhydrous form. 

4. Some references are missing in the discussion. In Line 157, the authors stated that "which was also observed by previous researchers against different anti-cancer drugs", but no reference was cited for these previous researchers.

Reviewer 2 Report

Development of stable liposomal drug delivery system of thy- 2 moquinone and its in-vitro anticancer studies using breast 3 cancer and cervical cancer cell lines is an interesting study; I have attached concerns regarding study design,

Major Comments:

No novelty exists in the current study; more importantly, the experiment design has many flaws.

Minor Comments:

1.       Why was the liposomes drug delivery system used instead of other drug delivery systems, e.g., nanosuspension, Nanoemulsion etc.?

2.       Define SiRNA[12], etc.

3.       Define CNS diseases and hepatotoxicity.

4.       Why is TEM size smaller than the Size measured through DLS?

5.       Why PDI is so high for both formulations?

6.       What about a DSC study for other ingredients for liposomal formulation?

7.       Why were different types of cell lines used in the study? Any scientific evidence?

8.       No statistical tools were applied for the current study.

Reviewer 3 Report

Comments to the Authors:

This is an interesting and overall well-written paper, this manuscript focuses on preparing thymoquinone (TQ)-loaded liposomes to improve their physico-chemical stability in gastric media and their performances in different cancer cell line studies. Compared with TQ, cell line studies show that TQ-loaded liposomal drug delivery systems have a better performance, which is greatly beneficial to the breast and cervical cancer treatment management. It will be a solid contribution to the Molecules and will certainly appeal to many of its readers. There are some issues should be addressed in the next few paragraphs. It is recommended that this manuscript be published in Molecules after completing revisions.

1.      There are a lot of grammar problems in the full text. please check and modify them. And please check the whole manuscript and correct the format issues. The writing of “℃” in the full text is not consistent.

2.      In section 2.2, the author stated that the average size of liposome is about 170~250 nm. However, the average size of liposome observed in Figure 1A is far below 100 nm. There is a contradiction between the text and picture. Moreover, Figure 1B is too vague to confirm the description of particle shape in 2.1: thymoquinone was possibly polygonal in shape. Please change a clearer picture.

3.      In Table 3, the difference in IC50 values between TQ-nano and TQ on BT549 is only 0.8. Such small difference doesn’t match the large gap in Figure 4. Please explain this inconsistency.

4.      In Figure 5, A24 and B24 are obviously clearer than A0 and B0, but authors did not explain this problem, please explain this phenomenon.

5.      The authors stated that TQ-nano shows increased cytotoxicity. Whether it is harmful to human bodies? Moreover, what are the advantages of this approach compared to other similar drugs?

6.      The introduction mainly introduces that the thymoquinone is used as an anti-cancer drug. In order to support this statement, the following recently published important related papers should be cited: Chem. Soc. Rev. 2017, 46, 7021; Chem. Soc. Rev. 2021, 50, 2839; Adv Mater. 2022, 34, 2106388.